# Simple Stopping Criteria for Information Theoretic Feature Selection

**DOI:** 10.3390/e21010099

**Published:** 2019-01-21

**Authors:** Shujian Yu, José C. Príncipe

**Affiliations:** Computational NeuroEngineering Laboratory, University of Florida, Gainesville, FL 32611, USA

**Keywords:** feature selection, stopping criterion, conditional mutual information, multivariate matrix-based Rényi’s α-entropy functional

## Abstract

Feature selection aims to select the smallest feature subset that yields the minimum generalization error. In the rich literature in feature selection, information theory-based approaches seek a subset of features such that the mutual information between the selected features and the class labels is maximized. Despite the simplicity of this objective, there still remain several open problems in optimization. These include, for example, the automatic determination of the optimal subset size (i.e., the number of features) or a stopping criterion if the greedy searching strategy is adopted. In this paper, we suggest two stopping criteria by just monitoring the conditional mutual information (CMI) among groups of variables. Using the recently developed multivariate matrix-based Rényi’s α-entropy functional, which can be directly estimated from data samples, we showed that the CMI among groups of variables can be easily computed without any decomposition or approximation, hence making our criteria easy to implement and seamlessly integrated into any existing information theoretic feature selection methods with a greedy search strategy.

## 1. Introduction

Feature selection aims to find the smallest feature subset that yields the minimum generalization error [1]. As a fundamental problem in machine learning and statistics communities, feature selection is relevant to understand better the classification problem at hand in many applications, such as medical decision making, economics, and engineering [2]. Consequently, it also has a large impact on interpretability or explainability, which is totally missing in most current deep learning techniques [3]. Ever since the pioneering work of Battiti [4], information theoretic feature selection has been extensively investigated in signal processing and machine learning communities (e.g., [5,6,7]). Given a set of *F* features $S=\{x_{1},x_{2},\cdots ,x_{F}\}$ (each $x_{i}$ denotes an attribute) and their corresponding class labels *y*, these methods seek a subset of informative attributes $S^{⋆}\subset S$, such that the mutual information (MI) between $S^{⋆}$ and *y* (i.e., $I(S^{⋆};y)$) is maximized [8].

Despite the simplicity of this objective, several open problems still remain in information theoretic feature selection. These include, for example, the reliable estimation of $I(S^{\prime };y)$ in high-dimensional space, in which $S^{\prime }$ denotes an arbitrary subset of *S* [8,9]. In fact, $S^{\prime }$ may contain both continuous and discrete variables, whereas *y* is a discrete variable. There is no universal agreement on the definition of MI between a discrete variable and a group of mixed variables, let alone its estimation [10]. Therefore, almost all existing information theoretic feature selection methods estimate $I(S^{\prime };y)$ by first discretizing the feature space and then approximating $I(S^{\prime };y)$ with low-order MI quantities, such as relevancy $I(x_{i};y)$, joint relevancy $I(\left\{x_{i},x_{j}\right\};y)$, conditional relevancy $I(x_{i};y|x_{j})$, redundancy $I(x_{i};x_{j})$, conditional redundancy $I(x_{i};x_{j}|y)$, and synergy [11]. These low-order MI quantities only capture the low-order feature dependency and, hence, severely limit the performance of existing information theoretic feature selection methods [12]. Interested readers can check Reference [8] for a systemic review of 17 popular low-order information theoretic criteria in the last two decades. Apart from MI estimation, another challenging problem, which also impacts performance greatly, is the automatic determination of the optimal size of $S^{⋆}$. This is because most of the information theoretic feature selection methods do not have a stopping criterion [1]. Hence, the user must find criteria to estimate the best number of features.

Regarding the first problem, our recent work [13] suggested that $I(S^{\prime };y)$ can be efficiently estimated using the normalized eigenspectrum of a Hermitian matrix of the projected data in the reproducing kernel Hilbert space (RKHS). In this paper, we expand on the topic of Reference [13] and illustrate that the novel multivariate matrix-based Rényi’s α-entropy functional also enables simple strategies to guide the early stopping in the greedy search procedure of information theoretic feature selection methods.

Before presenting our methods, we first briefly review related work and also point out a common issue in most previous methods. Perhaps the most acknowledged stopping criterion for information theoretic feature selection is that the value of $I(S^{\prime };y)$ stops increasing or reaches its maximum [14,15]. Given a new feature $x^{\prime }$, this rule suggests that we should stop selection if $I\left(\left\{S^{\prime },x^{\prime }\right\};y\right)\leq I\left(S^{\prime };y\right)$. An alternative approach is using the concept of the Markov blanket (MB) [16,17]. By definition, the MB *M* of a target variable *y* is the smallest subset of *S* such that *y* is conditional independent of the rest of the variables S−M, i.e., y⊥(S−M)|M [1]. From the perspective of information theory, this rule indicates that we should stop selection if the conditional mutual information (CMI) I({S−M};y|M) is zero.

Despite their simplicity, both rules are overoptimistic and cannot be applied in practice. In fact, by the chain rule of mutual information [18], we have:(1)$$I\left(\left\{S^{\prime },x^{\prime }\right\};y\right)=I\left(S^{\prime };y\right)+I\left(x^{\prime };y|S^{\prime }\right),$$
and:(2)I({S−M};y|M)=I(S;y)−I(M;y).

According to Equations (Equation 1) and (Equation 2), the feature selection is stopped if and only if the CMI item $I(x^{\prime };y|S^{\prime })$ or I({S−M};y|M) is zero. Unfortunately, since CMI is always non-negative [18] and rarely reduces to zero in practice due to statistical variation and chance agreement between variables [19], we always have $I\left(\left\{S^{\prime },x^{\prime }\right\};y\right)>I\left(S^{\prime };y\right)$ and I({S−M};y|M)>0. That is to say, the maximum value of $I(S^{\prime };y)$ is exactly I(S;y) and a perfect MB of *y* is perhaps the feature set *S* itself.

Admittedly, one can say that we can stop the selection if the increment of $I(S^{\prime };y)$ or the decrement of $I(\left\{S-S^{\prime }\right\};y|S^{\prime })$ approaches zero with a tiny residual. Unfortunately, since we still do not have a reliable estimator of MI and CMI in high-dimensional space (before [13]), it is hard for us to measure or determine how small the residual terms are.

To the best of our knowledge, there are only two methods in the literature that can stop the greedy search. François et al. [14] suggested monitoring the value of $I(S^{\prime };y)$ using a permutation test [20]. Specifically, suppose the new feature selected in the current iteration is $x^{\mathrm{cand}}$, the authors create a random permutation of $x^{\mathrm{cand}}$ (without permuting the corresponding *y*), denoted ${\tilde{x}}^{\mathrm{cand}}$. If $I(\left\{S^{\prime },x^{\mathrm{cand}}\right\};y)$ is not significantly larger than $I(\left\{S^{\prime },{\tilde{x}}^{\mathrm{cand}}\right\};y)$, $x^{\mathrm{cand}}$ can be discarded and the feature selection is stopped. Moreover, François et al. also suggested using the *k*-nearest neighbors (KNN) estimator [21] to estimate $I(S^{\prime };y)$. However, their results indicated that this estimator may result in negative CMI quantities no matter the selection of *k*. Vinh et al. [19], on the other hand, proposed monitoring the increment of $I(S^{\prime };y)$ after adding $x^{\mathrm{cand}}$ (i.e., $I({\tilde{x}}^{\mathrm{cand}};y|S^{\prime })$) using $\chi^{2}$ distribution. If $I(x^{\mathrm{cand}};y|S^{\prime })$ is smaller than a threshold obtained from the $\chi^{2}$ distribution at a certain significance level, the feature selection is stopped. However, one should note that the $\chi^{2}$ distribution assumption holds if and only if $I({\tilde{x}}^{\mathrm{cand}};y|S^{\prime })=0$ [19,22]. Unfortunately, as we emphasized earlier, the CMI quantity rarely reduces to zero in practice. This is also the reason why Vinh’s method is likely to severely underestimate the require number of features, as is illustrated in the experiments.

Different from previous work, we suggest using the novel multivariate matrix-based Rényi’s α-entropy functional [23] to estimate MI and CMI in high-dimensional space. We also present two simple stopping criteria based on these new estimators. To summarize, our main contributions are twofold:Efficient stopping criteria for feature selection are still missing in the literature. We present two rules based on the new estimators of information theoretic descriptors of Rényi’s α-entropy in RKHS. One should note that the properties, utilities, and possible applications of these new estimators are rather new and mostly unknown to practitioners;Our criteria are extremely simple and flexible. First, the proposed tests and stopping criteria can be incorporated into any sequential feature selection procedure, such as mutual information-based feature selection (MIFS) [4] and maximum-relevance minimum-redundancy (MRMR) [24]. Second, benefiting from the higher accuracy of the estimation, we demonstrated that one can simply use a threshold in a heuristic manner, which is very valuable for practitioners across different domains.

## 2. Simple Stopping Criteria for Information Theoretic Feature Selection

In this section, we start with a brief introduction to the recently proposed matrix-based Rényi’s α-entropy functional and its multivariate extension [13]. The novel definition yields two simple stopping criteria as presented below.

### 2.1. Matrix-Based Rényi’s α-Entropy Functional and Its Multivariate Extension

In information theory, a natural extension of the well-known Shannon’s entropy is Rényi’s α-order entropy [25]. For a random variable *X* with probability density function (PDF) f(x) in a finite set X, the α-entropy $H_{\alpha }\left(X\right)$ is defined as:
(3)$$H_{\alpha }\left(f\right)=\frac{1}{1-\alpha }\log\int_{X}f^{\alpha }\left(x\right)dx.$$

Based on this entropy definition, Rényi then proposed a divergence measure (α-relative entropy) between random variables with PDFs *f* and *g*:
(4)$$D_{\alpha }\left(f||g\right)=\frac{1}{\alpha -1}\log\int_{X}f\left(x\right){\left(\frac{f(x)}{g(x)}\right)}^{\alpha -1}dx.$$

Rényi’s entropy and divergence have a long track record of usefulness in information theory and its applications [26]. Unfortunately, the accurate PDF estimation impedes its more widespread adoption in data driven science. To solve this problem, References [13,23] suggest similar quantities that resemble quantum Rényi’s entropy [27] in terms of the normalized eigenspectrum of the Hermitian matrix of the projected data in RKHS, thus estimating the entropy and joint entropy among two or multiple variables directly from data without PDF estimation. For brevity, we directly give the definition.

**Definition** **1.**
*Let κ:X×X↦R be a real valued positive definite kernel that is also infinitely divisible [28]. Given $X=\{x^{1},x^{2},\ldots ,x^{n}\}$ and the Gram matrix K obtained from κ on all pairs of exemplars, that is, ${\left(K\right)}_{ij}=\kappa \left(x^{i},x^{j}\right)$, a matrix-based analogue to Rényi’s α-entropy for X can be defined, using a normalized positive definite (NPD) matrix A of size n×n and trace 1, by:*
(5)$$S_{\alpha }\left(A\right)=\frac{1}{1-\alpha }\log_{2}\left(\mathrm{tr}(A^{\alpha })\right)=\frac{1}{1-\alpha }\log_{2}\left[\sum_{i=1}^{n}\lambda_{i}{\left(A\right)}^{\alpha }\right],$$
*where $A_{ij}=\frac{1}{n}\frac{K_{ij}}{\sqrt{K_{ii}K_{jj}}}$ and $\lambda_{i}\left(A\right)$ denotes the i-th eigenvalue of A.*


The matrix functional in Equation (Equation 5) bears a lot of resemblance to well-known operational quantities from quantum information theory [27], where the density matrix (operator) ρ can be employed to compute expectation over an observable represented by the operator *A* as 〈A〉=tr(ρA). For instance, the quantum extensions of Rényi’s entropy [29,30] is given by:
(6)$$S_{\alpha }\left(\rho \right)=\frac{1}{1-\alpha }\log\left[\mathrm{tr}(\rho^{\alpha })\right]\;.$$

Although some properties of Equation (Equation 6) also apply to Equation (Equation 5), the matrix-based estimation is very different, since it deals with the Gram matrix obtained from pairwise evaluations of a positive definite kernel from data samples. Consequently, this novel definition not only involves the functional, but also the kernels employed to construct positive definite matrix.

**Definition** **2.**
*Given a collection of n samples ${\left\{s_{i}=\left(x_{1}^{i},x_{2}^{i},\cdots ,x_{k}^{i}\right)\right\}}_{i=1}^{n}$, where the superscript i denotes the sample index, each sample contains k (k≥2) measurements $x_{1}\in X_{1}$, $x_{2}\in X_{2}$, ⋯, $x_{k}\in X_{k}$ obtained from the same realization, and the positive definite kernels $\kappa_{1}:X_{1}\times X_{1}\mapsto R$, $\kappa_{2}:X_{2}\times X_{2}\mapsto R$, ⋯, $\kappa_{k}:X_{k}\times X_{k}\mapsto R$, let us denote $X_{l}=\left\{x_{l}^{1},x_{l}^{2},\cdots ,x_{l}^{n}\right\}\left(1\leq l\leq k\right)$ containing the l-th measurement of all samples, the Rényi’s α-order joint-entropy among k variables (denote here $J_{\alpha }\left(X_{1},X_{2},\cdots ,X_{k}\right)$) can be defined using the following matrix-based analogue:*
(7)$$J_{\alpha }\left(X_{1},X_{2},\cdots ,X_{k}\right)=S_{\alpha }\left(\frac{A_{1}⊙A_{2}⊙\cdots ⊙A_{k}}{\mathrm{tr}(A_{1}⊙A_{2}⊙\cdots ⊙A_{k})}\right),$$
*where ${\left(A_{1}\right)}_{ij}=\kappa_{1}\left(x_{1}^{i},x_{1}^{j}\right)$, ${\left(A_{2}\right)}_{ij}=\kappa_{2}\left(x_{2}^{i},x_{2}^{j}\right)$, ⋯, ${\left(A_{k}\right)}_{ij}=\kappa_{k}\left(x_{k}^{i},x_{k}^{j}\right)$, and *⊙* denotes the Hadamard product.*


The following two corollaries (proved in Reference [13]) serve as a foundation for our Definition 2. Specifically, Corollary 1 indicates that the joint entropy of a set of variables is greater than or equal to the maximum of all of the individual entropies of the variables in the set, whereas Corollary 2 suggests that the joint entropy of a set of variable is less than or equal to the sum of the individual entropies of the variables in the set.

**Corollary** **1.**
*Let [C] be the index set {1,2,⋯,C}. We partition [C] into two complementary subsets s and $\tilde{s}$. For any s⊂[C], denote all indices in s with $s_{1},s_{2},\cdots ,s_{\left|s\right|}$, where |·| stands for cardinality. Similarly, denote all indices in $\tilde{s}$ with ${\tilde{s}}_{1},{\tilde{s}}_{2},\cdots ,{\tilde{s}}_{|\tilde{s}|}$. Also let $A_{1}$, $A_{2}$, ⋯, $A_{C}$ be C n×n positive definite matrices with trace 1 and non-negative entries, and ${\left(A_{1}\right)}_{ii}={\left(A_{2}\right)}_{ii}=\cdots ={\left(A_{C}\right)}_{ii}=\frac{1}{n}$, for i=1,2,⋯,n. Then, the following two inequalities hold:*
(8)$$\begin{array}{c} S_{\alpha }\left(\frac{A_{1}⊙A_{2}⊙\cdots ⊙A_{C}}{\mathrm{tr}(A_{1}⊙A_{2}⊙\cdots ⊙A_{C})}\right)\leq S_{\alpha }\left(\frac{A_{s_{1}}⊙A_{s_{2}}⊙\cdots ⊙A_{s_{\left|s\right|}}}{\mathrm{tr}(A_{s_{1}}⊙A_{s_{2}}⊙\cdots ⊙A_{s_{\left|s\right|}})}\right)+S_{\alpha }\left(\frac{A_{{\tilde{s}}_{1}}⊙A_{{\tilde{s}}_{2}}⊙\cdots ⊙A_{{\tilde{s}}_{|\tilde{s}|}}}{\mathrm{tr}(A_{{\tilde{s}}_{1}}⊙A_{{\tilde{s}}_{2}}⊙\cdots ⊙A_{{\tilde{s}}_{|\tilde{s}|}})}\right), \end{array}$$
(9)$$\begin{array}{c} S_{\alpha }\left(\frac{A_{1}⊙A_{2}⊙\cdots ⊙A_{C}}{\mathrm{tr}(A_{1}⊙A_{2}⊙\cdots ⊙A_{C})}\right)\geq \max\left[S_{\alpha }\left(\frac{A_{s_{1}}⊙A_{s_{2}}⊙\cdots ⊙A_{s_{\left|s\right|}}}{\mathrm{tr}(A_{s_{1}}⊙A_{s_{2}}⊙\cdots ⊙A_{s_{\left|s\right|}})}\right),S_{\alpha }\left(\frac{A_{{\tilde{s}}_{1}}⊙A_{{\tilde{s}}_{2}}⊙\cdots ⊙A_{{\tilde{s}}_{|\tilde{s}|}}}{\mathrm{tr}(A_{{\tilde{s}}_{1}}⊙A_{{\tilde{s}}_{2}}⊙\cdots ⊙A_{{\tilde{s}}_{|\tilde{s}|}})}\right)\right]. \end{array}$$


**Corollary** **2.**
*Let $A_{1}$, $A_{2}$, ⋯, $A_{C}$ be C n×n positive definite matrices with trace *1* and non-negative entries, and ${\left(A_{1}\right)}_{ii}={\left(A_{2}\right)}_{ii}=\cdots ={\left(A_{C}\right)}_{ii}=\frac{1}{n}$, for i=1,2,⋯,n. Then, the following two inequalities hold:*
(10)$$S_{\alpha }\left(\frac{A_{1}⊙A_{2}⊙\cdots ⊙A_{C}}{\mathrm{tr}(A_{1}⊙A_{2}⊙\cdots ⊙A_{C})}\right)\leq S_{\alpha }\left(A_{1}\right)+S_{\alpha }\left(A_{2}\right)+\cdots +S_{\alpha }\left(A_{C}\right),$$
(11)$$S_{\alpha }\left(\frac{A_{1}⊙A_{2}⊙\cdots ⊙A_{C}}{\mathrm{tr}(A_{1}⊙A_{2}⊙\cdots ⊙A_{C})}\right)\geq \max\left[S_{\alpha }\left(A_{1}\right),S_{\alpha }\left(A_{2}\right),\cdots ,S_{\alpha }\left(A_{C}\right)\right].$$


### 2.2. Stopping Criteria Based on Conditional Mutual Information

Denote $\left\{x_{1},x_{2},\cdots ,x_{k}\right\}$, the selected features in $S^{\prime }$, $\left\{x_{1}^{\prime },x_{2}^{\prime },\cdots ,x_{m}^{\prime }\right\}$, the remaining features in $S-S^{\prime }$, given the entropy and joint entropy estimators shown in Equations (Equation 5)–(Equation 7), the MI between *y* and $S^{\prime }$ (i.e., $I(S^{\prime };y)$) and the CMI between *y* and $S-S^{\prime }$ conditioning on $S^{\prime }$ (i.e., $I(\left\{S-S^{\prime }\right\};y|S^{\prime })$), with analogue properties to Shannon’s definition of MI and CMI (by Shannon’s definition, $I\left(S^{\prime };y\right)=H\left(S^{\prime }\right)+H\left(y\right)-H\left(S^{\prime },y\right)$ and $I\left(\left\{S-S^{\prime }\right\};y|S^{\prime }\right)=H\left(S-S^{\prime },S^{\prime }\right)+H\left(y,S^{\prime }\right)-H\left(S-S^{\prime },y,S^{\prime }\right)-H\left(S^{\prime }\right)$, where H denotes entropy or joint entropy), which can be estimated with Equations (Equation 12) and (Equation 13), respectively, where $A_{1}$, $A_{2}$, ⋯, $A_{k}$, *B*, $C_{1}$, $C_{2}$, ⋯, $C_{m}$ denote the Gram matrices evaluated over $x_{1}$, $x_{2}$, ⋯, $x_{k}$, *y*, $x_{1}^{\prime }$, $x_{2}^{\prime }$, ⋯, $x_{m}^{\prime }$, respectively, and ⊙ denotes the Hadamard product.

(12)$$I_{\alpha }\left(B;\left\{A_{1},A_{2},\cdots ,A_{k}\right\}\right)=S_{\alpha }\left(B\right)+S_{\alpha }\left(\frac{A_{1}⊙A_{2}⊙\cdots ⊙A_{k}}{\mathrm{tr}(A_{1}⊙A_{2}⊙\cdots ⊙A_{k})}\right)-S_{\alpha }\left(\frac{A_{1}⊙A_{2}⊙\cdots ⊙A_{k}⊙B}{\mathrm{tr}(A_{1}⊙A_{2}⊙\cdots ⊙A_{k}⊙B)}\right),$$

(13)$$\begin{array}{cc} I_{\alpha }\left(\left\{C_{1},C_{2},\cdots ,C_{m}\right\};B|\left\{A_{1},A_{2},\cdots ,A_{k}\right\}\right) & =S_{\alpha }\left(\frac{A_{1}⊙A_{2}⊙\cdots ⊙A_{k}⊙C_{1}⊙C_{2}⊙\cdots ⊙C_{m}}{\mathrm{tr}(A_{1}⊙A_{2}⊙\cdots ⊙A_{k}⊙C_{1}⊙C_{2}⊙\cdots ⊙C_{m})}\right)\\ & +S_{\alpha }\left(\frac{B⊙A_{1}⊙A_{2}⊙\cdots ⊙A_{k}}{\mathrm{tr}(B⊙A_{1}⊙A_{2}⊙\cdots ⊙A_{k})}\right)\\ & -S_{\alpha }\left(\frac{A_{1}⊙A_{2}⊙\cdots ⊙A_{k}⊙B⊙C_{1}⊙C_{2}⊙\cdots ⊙C_{m}}{\mathrm{tr}(A_{1}⊙A_{2}⊙\cdots ⊙A_{k}⊙B⊙C_{1}⊙C_{2}⊙\cdots ⊙C_{m})}\right)-S_{\alpha }\left(\frac{A_{1}⊙A_{2}⊙\cdots ⊙A_{k}}{\mathrm{tr}(A_{1}⊙A_{2}⊙\cdots ⊙A_{k})}\right). \end{array}$$

As can be seen, the multivariate matrix-based Rényi’s α-entropy functional enables simple estimation of both MI and CMI in high-dimensional space, no matter the data characteristics (e.g., continuous or discrete) in each dimension. In contrast to previous definitions on Rényi’s α-mutual information [31] that stem from a probabilistic perspective using expectations over PDFs, Equations  (Equation 12) and (Equation 13) directly estimate MI and CMI in RKHS. Benefiting from these elegant expressions, suppose the new feature selected in the current iteration is $x^{\mathrm{cand}}$, we present two simple criteria to guide the early stopping of the greedy search. Specifically, we aim to test the “goodness-of-fit” of the MB condition, i.e., $S-S^{\prime }$ is the MB of *y* given $S^{\prime }$. Intuitively, if $I(\left\{S-S^{\prime }-x^{\mathrm{cand}}\right\};y|\left\{S^{\prime },x^{\mathrm{cand}}\right\})$ approaches zero, the MB condition is approximately satisfied.

**Criterion** **I.**If $I(\left\{S-S^{\prime }-x^{\mathrm{cand}}\right\};y|\left\{S^{\prime },x^{\mathrm{cand}}\right\})\leq \varepsilon$, where ε refers to a tiny threshold, then we should stop the selection. We term this criterion CMI-heuristic, since ε is a heuristic value.

**Criterion** **II.**Motivated by Reference [14], in order to quantify how $x^{\mathrm{cand}}$ affects the MB condition, we created a random permutation of $x^{\mathrm{cand}}$ (without permuting the corresponding *y*), denoted ${\tilde{x}}^{\mathrm{cand}}$. If $I(\left\{S-S^{\prime }-x^{\mathrm{cand}}\right\};y|\left\{S^{\prime },x^{\mathrm{cand}}\right\})$ is not significantly smaller than $I(\left\{S-S^{\prime }-{\tilde{x}}^{\mathrm{cand}}\right\};y|\left\{S^{\prime },{\tilde{x}}^{\mathrm{cand}}\right\})$, $x^{\mathrm{cand}}$ can be discarded and the feature selection is stopped. We term this criterion CMI-permutation (see Algorithm 1 for more details, where 1 is an indicator function which is 1 if its argument is true and 0 otherwise.).

**Algorithm 1** CMI-permutation**Input:** Feature set *S*; Selected feature subset S′; Class labels *y*; Selected feature in the current iteration xcand; Permutation number *P*; Significance level θ.**Output:**
*decision* (Stop selection or Continue selection).  1:Estimate I({S−S′−xcand};y|{S′,xcand}) with Equation (Equation 13).  2:**for**i=1 to *P*
**do**  3:    Randomly permute xcand to obtain x˜icand.  4:    Estimate I({S−S′−x˜icand};y|{S′,x˜icand}) with Equation (Equation 13).  5:**end for**  6:**if**∑i=1P1[I({S−S′−xcand};y|{S′,xcand})≥I({S−S′−x˜icand};y|{S′,x˜icand})]P≤θ**then**  7:    *decision*←Continue selection.  8:**else**  9:    *decision*←Stop selection.10:**end if**11:**return***decision*

## 3. Experiments and Discussions

We compared our two criteria with existing ones [14,19] on 10 well-known public datasets used in previous feature selection research [8,19], covering a wide variety of sample-feature ratios and a range of multiclass problems. The detailed properties of these datasets, including the number of features (*F*), the number of examples (*N*), and the number of classes (*C*), are available in Reference [8]. We refer the criterion in Reference [19] ΔMI-$\chi^{2}$, since it monitors the increment of MI $I(S^{\prime };y)$ (i.e., $I(x^{\mathrm{cand}};y|S^{\prime })$) with $\chi^{2}$ distribution. We refer the criterion in Reference [14] MI-permutation, since it uses the permutation test to quantify the impact of $x^{\mathrm{cand}}$ on $I(S^{\prime };y)$. Throughout this paper, we selected $\varepsilon =10^{-4}$ in CMI-heuristic and used the multivariate matrix-based Rényi’s α-entropy functional to estimate all MI quantities in MI-permutation. The baseline information feature selection method used in this paper is from Reference [13], which directly optimizes $I(S^{\prime };y)$ in a greedy manner without any decomposition or approximation. An example for different stopping criteria on the dataset waveform is shown in Figure 1.

To provide a comprehensive evaluation, we tested the performance of the novel Rényi’s α-entropy estimator with α=1.01 (to approximate Shannon’s definition) and α=2 (i.e., the quadratic information quantities that have been widely applied in information theoretic learning [26]). Moreover, we also tested the performance of different statistical tests with significance level θ=0.95 and θ=0.99, but since the results are very similar, herein, we only report results with θ=0.95. The quantitative results, under different values of α, are summarized in Table 1 and Table 2. For each criterion, we reported the number of selected features and the average classification accuracy across 100 bootstrap runs. In each run, *N* bootstrap samples are drawn for the training set, while the unselected samples serve as the test set. Same as in Reference [8], we used the linear support vector machine (SVM) [32] as the baseline classifier. To give a reference, we defined the “optimal” number of features (an unknown parameter) as the one that yields the maximum bootstrap accuracy or first achieves a bootstrap accuracy with no statistical difference to the maximum value (evaluated by paired t-test with significance level 0.05), and ranked all the criteria based on the difference between their estimated number of features and the optimal one.

As can be seen, ΔMI-$\chi^{2}$ is likely to severely underestimate the number of features, accompanied by the lowest bootstrap accuracy. One possible reason is that $I(x^{\mathrm{cand}};y|S^{\prime })$ does not precisely fit a $\chi^{2}$ distribution if the MB condition is not satisfied. CMI-permutation and MI-permutation always have the same ranks. This is because $I\left(S^{\prime };y\right)+I\left(\left\{S-S^{\prime }\right\};y|S^{\prime }\right)=I\left(S;y\right)$, a fixed value. Thus, it is equivalent to monitor the increment of $I(S^{\prime };y)$ or the decrement of $I(\left\{S-S^{\prime }\right\};y|S^{\prime })$. On the other hand, it is surprising to find that CMI-heuristic performs the best in most datasets. This indicates that although the permutation test is effective to test the MB condition, $\varepsilon =10^{-4}$ is a reliable threshold to speed up this test, as the permutation test is always time-consuming. If we look deeper, it is also interesting to see that different values of α in Rényi’s entropy affect the results. This is to be expected, because the α is associated with the norm selected in the simplex to find the distance of the PDF to the center of the space. Effectively. different α weights change the effects of the tails, as explained in Reference [26]: Values of α smaller than 1 emphasize the importance of samples in the tails, values larger than 2 emphasize the modes of the distributions, whereas values close to 2 give equal weights to every sample, no matter where it is located in the distribution. Since classification uses a counting norm, values of α lower than 2 should be preferred. This is also the reason CMI-permutation and MI-permutation perform better with α=1.01. Finally, the Wilcoxon rank-sum test at 0.1 significance level, shown in Table 3, corroborates our analysis that our criteria perform equally to or slightly better than MI-permutation, but significantly better than ΔMI-$\chi^{2}$.

## 4. Conclusions

This paper suggests two simple stopping criteria, namely, CMI-heuristic and CMI-permutation, for information theoretic feature selection by monitoring the value of CMI estimated with the novel multivariate matrix-based Rényi’s α-entropy functional. Experiments on 10 benchmark datasets indicate that (1) CMI is a more tractable quantity than MI, and (2) the non-parametric test (like the permutation test) is more effective than the parametric test with a prior distribution (like the $\chi^{2}$ distribution), to guide early stopping in feature selection. Moreover, as an alternative to a permutation test, we also demonstrate a tiny threshold is sufficient to test the MB condition, which is very valuable for practitioners across different domains.

In the future, we will investigate the performance of our stopping criteria on big data. This is because with the tremendous growth of dataset sizes and dimensionality, the scalability of most feature selection methods could be jeopardized [33]. This problem is particularly important for our criteria, as the large-scale eigenvalue decomposition is always a challenging problem mathematically [34]. Moreover, we are also interested in testing our criteria on different types of data, including the linked data that is pervasive in social media and bioinformatics [33].

## Figures and Tables

**Figure 1 entropy-21-00099-f001:**
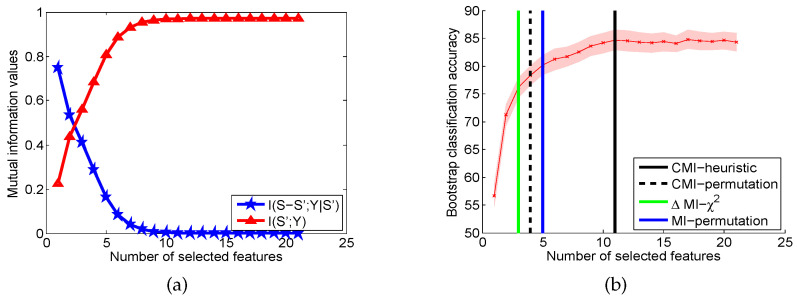
(**a**) shows the the values of mutual information (MI) $I(S^{\prime };y)$ and conditional mutual information (CMI) $I(\left\{S-S^{\prime }\right\};y|S^{\prime })$ with respect to different number of selected features, i.e., the size of $S^{\prime }$. $I(S^{\prime };y)$ is monotonically increasing, whereas $I(\left\{S-S^{\prime }\right\};y|S^{\prime })$ is monotonically decreasing. (**b**) shows the terminated points produced by different stopping criteria, namely CMI-heuristic (black solid line), CMI-permutation (black dashed line), ΔMI-$\chi^{2}$ (green solid line), and MI-permutation (blue solid line). The red curve with the shaded area indicates the average bootstrap classification accuracy with 95% confidence interval. In this example, the bootstrap classification accuracy reaches its statistical maximum value with 11 features and CMI-heuristic performs the best.

**Table 1 entropy-21-00099-t001:** The number of selected features (#F) and the bootstrap classification accuracy (acc.) comparison for CMI-heuristic and CMI-permutation against different stopping criteria. We set α=1.01 for Rényi’s α-entropy functional. All criteria are ranked based on the difference between their selected number of features and the optimal values. The best two ranks are marked with green and blue, respectively. The average rank across all datasets is reported in the bottom line. The value behind the name of each dataset indicates the total number of features.

	CMI-Heuristic			CMI-Permutation			ΔMI-$\chi^{2}$ [19]			MI-Permutation [14]			“Optimal”	
	#F	acc.	rank	#F	acc.	rank	#F	acc.	rank	#F	acc.	rank	#F	acc.
waveform (21)	11	84.7 ± 1.8	1	4	78.3 ± 1.8	3	3	76.1 ± 1.8	4	5	80.2 ± 1.8	2	11	84.7 ± 1.8
breast (30)	2	92.3 ± 1.7	1	2	92.3 ± 1.7	1	2	92.3 ± 1.7	1	2	92.3 ± 1.7	1	2	95.2 ± 1.2
heart (13)	13	81.7 ± 3.5	4	4	80.4 ± 3.3	1	2	76.9 ± 3.8	3	4	80.4 ± 3.3	1	6	82.6 ± 3.0
spect (22)	22	80.6 ± 3.6	4	11	82.1 ± 3.3	1	1	80.1 ± 3.3	3	7	81.1 ± 3.3	2	11	82.1 ± 3.3
ionosphere (34)	15	83.3 ± 2.8	1	7	81.8 ± 2.8	2	1	76.7 ± 3.2	4	7	81.8 ± 2.8	2	33	85.3 ± 3.0
parkinsons (22)	12	85.2 ± 3.7	1	4	85.0 ± 3.2	2	1	85.1 ± 3.5	4	4	85.0 ± 3.2	2	9	86.5 ± 3.4
semeion (256)	59	86.1 ± 1.3	1	20	77.7 ± 1.5	2	4	49.6 ± 1.7	4	20	77.7 ± 1.5	2	73	93.3 ± 1.3
Lung (325)	5	74.2 ± 7.7	3	10	73.9 ± 8.0	2	1	46.5 ± 7.5	4	13	79.1 ± 7.9	1	41	84.3 ± 6.5
Lympth (4026)	6	81.3 ± 5.8	1	248	88.7 ± 6.1	3	2	62.8 ± 6.5	2	249	88.9 ± 6.2	4	70	90.7 ± 5.4
Madelon (500)	3	69.5 ± 1.6	2	2	59.5 ± 1.6	3	4	76.7 ± 1.5	1	2	59.5 ± 1.6	3	4	76.7 ± 1.5
average rank			1.9			2.0			3.0			2.0		

**Table 2 entropy-21-00099-t002:** The number of selected features (#F) and the bootstrap classification accuracy (acc.) comparison for CMI-heuristic and CMI-permutation against different stopping criteria. We set α=2 for Rényi’s α-entropy functional. All criteria are ranked based on the difference between their selected number of features and the optimal values. The best two ranks are marked with green and blue, respectively. The average rank across all datasets is reported in the bottom line. The value behind the name of each dataset indicates the total number of features.

	CMI-Heuristic			CMI-Permutation			ΔMI-$\chi^{2}$ [19]			MI-Permutation [14]			“Optimal”	
	#F	acc.	rank	#F	acc.	rank	#F	acc.	rank	#F	acc.	rank	#F	acc.
waveform (21)	17	84.2 ± 1.9	1	4	78.2 ± 2.1	2	3	76.1 ± 2.0	4	4	78.2 ± 2.1	2	11	84.7 ± 1.8
breast (30)	1	90.9 ± 1.5	2	1	90.9 ± 1.5	2	2	95.2 ± 1.2	1	1	90.9 ± 1.5	2	2	95.2 ± 1.2
heart (13)	6	82.6 ± 3.0	1	1	55.0 ± 4.4	3	2	76.9 ± 3.8	2	1	55.0 ± 4.4	3	6	82.6 ± 3.0
spect (22)	17	80.2 ± 4.2	1	2	78.2 ± 3.2	3	1	79.6 ± 3.2	4	3	79.1 ± 3.2	2	11	81.4 ± 4.2
ionosphere (34)	18	84.7 ± 2.9	1	8	81.9 ± 3.5	2	1	76.4 ± 3.9	4	8	81.9 ± 3.5	2	33	85.3 ± 3.0
parkinsons (22)	13	83.7 ± 3.7	1	4	84.6 ± 3.2	2	1	77.0 ± 5.0	4	3	84.6 ± 3.4	3	9	86.0 ± 3.9
semeion (256)	53	85.5 ± 1.1	1	40	82.7 ± 1.4	2	4	49.6 ± 1.7	4	40	82.7 ± 1.4	2	73	93.3 ± 1.3
Lung (325)	9	76.1 ± 7.9	2	5	74.2 ± 7.7	3	1	46.5 ± 7.5	4	11	73.0 ± 7.1	1	41	84.3 ± 6.5
Lympth (4026)	8	86.0 ± 5.3	3	65	89.6 ± 5.5	1	2	62.8 ± 6.5	4	64	89.6 ± 5.5	2	70	90.7 ± 5.4
Madelon (500)	3	69.5 ± 1.6	2	1	52.4 ± 1.7	3	4	76.7 ± 1.5	1	1	52.4 ± 1.7	3	4	76.7 ± 1.5
average rank			1.5			2.3			3.2			2.2		

**Table 3 entropy-21-00099-t003:** Summary of *p*-values and decisions (in parentheses) of Wilcoxon rank-sum test at 0.1 significance level on ranks of our criteria against ΔMI-$\chi^{2}$ and MI-permutation. A *p*-value smaller than 0.1 indicates rejection of the null hypothesis that two criteria perform equally.

α=1.01		
	Δ **MI-** $\chi^{2}$	**MI-Permutation**
CMI-heuristic	0.0781 (1)	0.5455 (0)
CMI-permutation	0.0561 (1)	0.9036 (0)
α=2		
	Δ **MI-** $\chi^{2}$	**MI-Permutation**
CMI-heuristic	0.0081 (1)	0.0341 (1)
CMI-permutation	0.0587 (1)	0.7340 (0)

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
