# Peer review of "Simple Stopping Criteria for Information Theoretic Feature Selection"

_entropy, 2019, doi:10.3390/e21010099_

Round 1

Reviewer 1 Report

This paper proposes to leverage recent methods to estimate Shannon’s mutual information to address issues of feature selection in the general context of data analysis. Although the paper is well written, I have questions regarding the relevance of the topic and the novelty of the contribution. In the following, I provide comments on these two issues, and then list some minor observations.

Major comments:

1. Many people in the machine learning community believe that the problem of feature selection has been effectively solved by deep neural networks. In effect, deep learning does not need to reduce the dimension of the input data, as the learning itself chooses the relevant features automatically. This being said, traditional feature selection still can be an issue for specific applications, where -for various reasons- deep learning cannot be applied. Given the this context, it is crucial for the authors to clarify why the problem they are tackling is still relevant.

2. It is not clear which is the contribution of the paper. On the one hand, the two proposed stopping criterion are very similar to existent ones. On the other hand, the proof that the method for estimating the mutual information based on equation (7) has already been developed in Ref. [11]. In fact, the actual use of the Renyi entropy is rather limited, as the proposed scheme up only uses alpha=1.

In summary, please clarify that the novelty of the current article is ok for being included in a well-established journal such as Entropy.

Minor observations:

3. While the abstract says that feature selection is about maximizing mutual information, the introduction says is about minimizing generalization error. Please note that these two goals are actually not equivalent. Moreover, the goal of maximizing mutual information trivializes feature selection, as one should always pick all the features.

4. In the section of related work, the observations related to Eqs. (1) and (2) are basically the same. It sounds weird and confusing to see the same arguments exposed twice; maybe merging these observations could improve the presentation.

5. The last methods presented in Section 1.1 seem good, and not so different from the ones proposed in this article. Please clarify if they have any weaknesses that are improved in the approach proposed in the current manuscript. 

6. Definitions 1 and 2 come out of the blue. It is not clear how kernels come into the picture, and how (5) is related to the Renyi entropy. It is very hard for the reader to get all these new things packed into these definitions. Please introduce better the ideas, and maybe provide an example.

7. Why S_\alpha is a function of A, and not of the actual variables? The way it is now, it seems a rather straightforward functional over the matrices, not necessarily related to Renyi divergences.

8. The notation with Hadamard products looks incovenient, specially for example in Eq. (8). Please consider introducing a different notation for all that.

9. Does Eq. (5) and (6) don’t depend on the kernel k(x,y)? 

10. It is not clear if Eqs. (7) and (8) are a definition, or something that can be proved. Please clarify this.

11. How does Eq. (7) relates to the existent definitions of mutual information based on Renyi entropy? See e.g.

Verdú, Sergio. "α-mutual information." In Information Theory and Applications Workshop (ITA), 2015, pp. 1-6. IEEE, 2015.

12. The alpha of the permutation test might induce confusion with the alpha of the Renyi’s entropies. 

Author Response

Please see attached file our reply.

Reviewer 2 Report

The problem discussed in this paper is how to find a samallest feature subset that yields the minimum generalization error.

If we have a set of $F$ features $S = \{x_1, x_2, \dots, x_F\}$ and the corresponding class labels $y$, the problem is

to find a subset of informative attributes $S^*$ of $S$ such that the mutual information (MI) between

$S^*$ and $y$ is maximized. In the present paper, the authors suggest two simple stopping criteria.

These criteria are developed using the matrix-based R\'enyi´s $\alpha$-entropy functional.

The results seem to be correct and are relevant for information theory. So I recommend the

publication of this paper in {\it Entropy}.

Author Response

We appreciate the positive comments from the reviewer. In our revised manuscript, we have improved our English expression and conducted more experiments to further strengthen the results. All the changes are marked with red in the revised manuscript.

Reviewer 3 Report

This paper suggests two stopping criteria for feature selection using conditional mutual information. It uses 10 data sets to compare the number of the selected criteria and the accuracy of classification with existing stopping criteria.

The paper suggests two stopping criteria CMI-heuristic and CMI-permutation, based on conditional mutual information. It compares the experimental resutls with ΔΜΙ-χ2 and MI-permutation, which bases the stopping criteria on permutation entropy.

The paper is well written; however I also find some weak points. My main concerns about this paper are:

a) the contribution of the paper is not so significant to be included in a journal at the scientific level of Entropy. I would expect to see a paper investigating stopping criteria published in a conference

b) the experimental results do not show a clear gain in the accuracy achieved. The authors invest on CMI-permutation. However, CMI-permutation and MI-permutation present similar results. The authors comment on it and explain it theoretically. The CMI-heuristic seem to present better results, but weak (according to statistical significance). 

c) I assume the accuracy and the number of the selected feature depends on the value of the thresholds selected for each criterion. However, no study on the values of the thresholds is presented or discussed. 

Minor:

- no need to separate subsection 1.1 from the rest of the introduction, since there is no sebsection 1.2. We usually do not split the introductory section into subsections. Especially here I see no reason to do it.

- why don't the authors use three lines for equation (8)?

- the authors call the first criterion CMI-heuristic due to the need to have a threshold value e. But all those methods need to define such a threshold, including CMI-permutation and CM-permutation

 - Algorithm 1, line 6: please allow some space so that it is easier to read

- please correct the quotation mark at the word Optimal (two times in page 5)

- please remove (CMI) at line 132. It has been defined before

- please add MI at the abbreviations

Author Response

Please see attached file our reply.

Round 2

Reviewer 1 Report

The authors have done well answering my observations. At this stage I only minor comments:

The notation S_\alpha(something) has three uses: eq. (5), lhs of (7) and rhs of (7). Please avoid overloading the same symbol with so many uses.

Is there any shorthand notation that could be used for the Hadamard product of a number of matrices?

I still find the notation in Eq. (7) a bit confusing, because although the quantity is a function of the samples and the kernel, the argument of S_\alpha is neither of them but some matrices...

I find the section of conclusions rather poor. Please try to give an account of the meaning of these results in a more general context, and discuss possible consequences and future work.

Author Response

We highly appreciate your valuable suggestions, which led to substantial improvements of the paper. Please see the attached file for our response.

Reviewer 3 Report

The reviewer thinks that the authors have sufficiently replied to the reviewer's comments/suggestions and also thinks that the paper is in a good scientific level.

Author Response

Thanks for your positive comments on our manuscript and response. We highly appreciate your 

valuable suggestions, which led to substantial improvements of the paper.